# Self-Verification Improves Few-Shot Clinical Information Extraction

**Zelalem Gero** [* 1]   **Chandan Singh** [* 1]   **Hao Cheng** [1]   **Tristan Naumann** [1]
**Michel Galley** [1]   **Jianfeng Gao** [1]   **Hoifung Poon** [1]

## Abstract

Extracting patient information from unstructured text is a critical task in health decision-support and clinical research. Large language models (LLMs) have shown the potential to accelerate clinical curation via few-shot in-context learning, in contrast to supervised learning, which requires costly human annotations. However, despite drastic advances, modern LLMs such as GPT-4 still struggle with issues regarding accuracy and interpretability, especially in safety-critical domains such as health. We explore a general mitigation framework using self-verification, which leverages the LLM to provide provenance for its own extraction and check its own outputs. This framework is made possible by the asymmetry between verification and generation, where the former is often much easier than the latter. Experimental results show that our method consistently improves accuracy for various LLMs across standard clinical information extraction tasks. Additionally, self-verification yields interpretations in the form of a short text span corresponding to each output, which makes it efficient for human experts to audit the results, paving the way towards trustworthy extraction of clinical information in resource-constrained scenarios. To facilitate future research in this direction, we release our code and prompts. [1]

## 1. Introduction and related work

Clinical information extraction plays a pivotal role in the analysis of medical records and enables healthcare practitioners to efficiently access and utilize patient data (Zweigen-baum et al., 2007; Wang et al., 2018). Few-shot learning approaches have emerged as a promising solution to tackle the scarcity of labeled training data in clinical information extraction tasks (Agrawal et al., 2022; Laursen et al., 2023). However, these methods continue to struggle with accuracy and interpretability, both critical concerns in the medical domain (Gutiérrez et al., 2022).

Here, we address these issues by using self-verification (SV) to improve few-shot clinical information extraction. SV builds on recent works that chain together large language model (LLM) calls to improve an LLM's performance (Wu et al., 2022; Wang et al., 2022; Chase, 2023). Intuitively, these chains succeed because an LLM may be able to perform individual steps in a task, *e.g.* evidence verification, more accurately than the LLM can perform an entire task, *e.g.* information extraction (Ma et al., 2023; Madaan et al., 2023; Zhang et al., 2023). Such chains have been successful in settings such as multi-hop question answering (Press et al., 2022), retrieval-augmented/tool-augmented question answering (Peng et al., 2023; Paranjape et al., 2023; Schick et al., 2023; Gao et al., 2023), and code execution (Jojic et al., 2023). Here, we analyze whether building such a chain can improve clinical information extraction.

Fig. 1 shows the SV pipeline we build here. We broadly define self-verification as using multiple calls to the *same* LLM to verify its output, and also to ground each element of its output in evidence. Our SV pipeline consists of four steps, each of which calls the same LLM with different prompts. First, the *Original extraction* step queries the LLM directly for the desired information. Next, the *Omission* step finds missing elements in the output, the *Evidence* step grounds each element in the output to a text span in the input, and the *Prune* step removes inaccurate elements in the output. Taken together, we demonstrate that these steps improve the reliability of extracted information.

Additionally, SV provides interpretable grounding for each output, in the form of a short text span in the input. Interpretability has taken many forms in NLP, including posthoc feature importance (Lundberg & Lee, 2017; Ribeiro et al., 2016), intrinsically interpretable models (Rudin, 2019; Tan et al., 2022; Singh et al., 2022a), and visualizing model intermediates, *e.g.* attention (Wiegreffe & Pinter, 2019). The

[*]Equal contribution [1]Microsoft Research. Correspondence to: Zelalem Gero <zelalemgero@microsoft.com>, Chandan Singh <chansingh@microsoft.com>.

*Workshop on Interpretable ML in Healthcare at International Conference on Machine Learning (ICML)*, Honolulu, Hawaii, USA. 2023. Copyright 2023 by the author(s).

[1]All code is made available at ⊙ github.com/microsoft/clinical-self-verification.

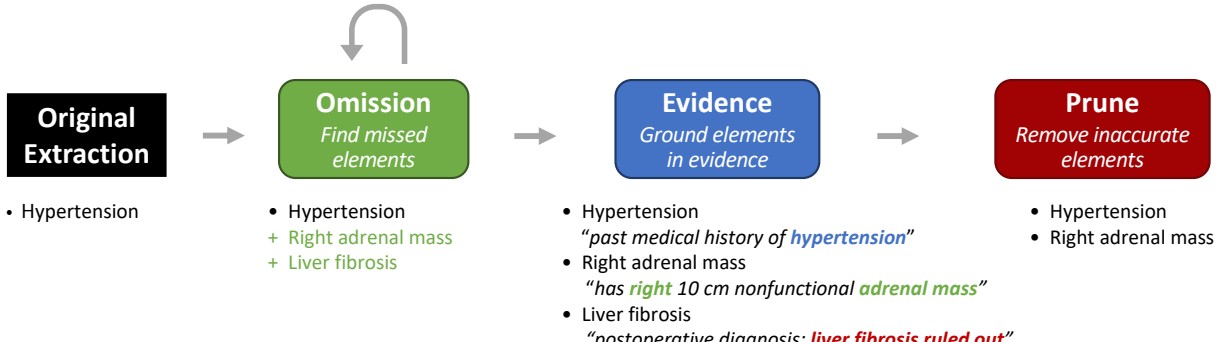

*Figure 1.* Overview of self-verification pipeline for clinical information extraction. Each step calls the same LLM with different prompts to refine the information from the previous steps. Below each step we show abbreviated outputs for extracting a list of assigned diagnoses from a sample clinical note.

interpretable grounding we generate comes directly from an LLM, similar to recent works that use LLMs to generate explanations (Rajani et al., 2019; MacNeil et al., 2022; Singh et al., 2023) and ground those explanations in evidence (Rashkin et al., 2021; Gao et al., 2022; Xue et al., 2023).

Experiments on various clinical information extraction tasks and various LLMs, including ChatGPT (GPT-4) (OpenAI, 2023) and ChatGPT (GPT-3.5) (Ouyang et al., 2022), show the efficacy of SV. In addition to improving accuracy, we find that the extracted interpretations match human judgements of relevant information, enabling auditing by a human and helping to build a path towards trustworthy extraction of clinical information in resource-constrained scenarios.

## 2. Methods and experimental setup

### 2.1. Methods: Self-verification

Fig. 1 shows the four different steps of the introduced SV pipeline. The pipeline takes in a raw text input, *e.g.* a clinical note, and outputs information in a pre-specified format, *e.g.* a bulleted list. It consists of four steps, each of which calls the same LLM with different prompts in order to refine and ground the original output.

The original extraction step uses a task-specific prompt which instructs the model to output a variable-length bulleted list. In the toy example in Fig. 1, the goal is to identify the two diagnoses *Hypertension* and *Right adrenal mass*, but the original extraction step finds only *Hypertension*.

After the original LLM extraction, the Omission step finds missing elements in the output; in the Fig. 1 example it finds *Right adrenal mass* and *Liver fibrosis*. For tasks with long inputs (mean input length greater than 2,000 characters), we repeat the omission step to find more potential missed elements (we repeat five times, and continue repeating until the omission step stops finding new omissions).

Next, the Evidence step grounds each element in the output to a text span in the input. The grounding in this step provides interpretations that can be inspected by a human. In the Fig. 1 example, we find quotes supporting the first two diagnoses, but the quote for *liver fibrosis* shows that it was in fact *ruled out*, and is therefore an incorrect diagnosis. Finally, the Prune step uses the supplied evidence to remove inaccurate elements from the output. In Fig. 1 this results in removing *liver fibrosis* to return the correct final list. Taken together, these steps help to extract accurate and interpretable information, with the omission step improving recall, the evidence step grounding findings, and the pruning step improving precision.

We provide the exact prompts used in all steps in the Github repo. For the tasks with short inputs, we include 5 random data demonstrations in the original extraction prompt; otherwise all prompts are fixed across examples. Prompts are straightforward instructions that specify the desired output format, e.g. (i) *Create a bulleted list of which medications are mentioned and whether they are active, discontinued, or neither*, or (ii) *List all medication names in the patient note that were missed in Extracted medications list. Put the medication name in quotes and the status in parentheses (active, discontinued, or neither). If no additional medications are found, return "None"*.

### 2.2. Experimental setup

**Datasets** Table 1 gives the details of each task we study here. Each task requires extracting a variable-length list of elements. In clinical trial arm extraction, these are names of different clinical trial arms, manually annotated from the EBM-NLP dataset (Nye et al., 2018). In the medication status extraction task, in addition to medication names the medication status must additionally be classified as *active*, *discontinued*, or *neither*. The text inputs for arm extraction / medication status extraction are relatively small (average length is 1,620 characters and 382 characters, respectively).

In the case of MIMIC-III and MIMIC-IV (Johnson et al., 2016; 2021), we predict ICD-9 or ICD-10 codes (corresponding to diagnoses and procedures). We predict ICD codes using relevant sections from all types of clinical notes for MIMIC-III (average length: 5,200 words) but only discharge summaries for MIMIC-IV (average length: 1,400 words). The ICD codes are not directly present in the text input, and therefore the task requires translating the diagnoses to their relevant code. MIMIC data is preprocessed using a standard pipeline (see Appendix A.1) and we evaluate on a random subset of 250 inputs for each task.

**Models** We evaluate three different models: GPT-3.5 (Brown et al., 2020), `text-davinci-003`, ChatGPT (GPT-3.5) (Ouyang et al., 2022) `gpt-3.5-turbo`, and ChatGPT (GPT-4) (OpenAI, 2023) `gpt4-0314` (in chat mode), all accessed securely through the Azure OpenAI API. We set the sampling temperature for LLM decoding to 0.1.

**Evaluation** Extraction is evaluated via case-insensitive exact string matching, and we report the resulting macro F1 scores, recall, and precision. In some cases, this evaluation may underestimate actual performance as a result of the presence of acronyms or different names within the output; nevertheless, the relative performance of different models/methods should still be preserved. Following common practice, we restrict ICD code evaluation to the top 50 codes appearing in the dataset.

## 3. Results

### 3.1. Self-verification improves prediction performance

Table 2 shows the results for clinical extraction performance with and without self-verification. Across different models and tasks, SV consistently provides a performance improvement. The performance improvement is occasionally quite large (*e.g.* ChatGPT (GPT-4) shows more than a 0.1 improvement in F1 for clinical trial arm extraction and more than a 0.3 improvement for medication status extraction), and the average F1 improvement across models and tasks is 0.056. We also compare to a baseline where we concatenate the prompts across different steps into a single large prompt which is then used to make a single LLM call for information extraction. We find that this large-prompt baseline performs slightly worse than the baseline reported in Table 2, which uses a straightforward prompt for extraction (see comparison details in Table A5).

For tasks with short inputs, we find that GPT-3.5 performs best, even outperforming ChatGPT (GPT-4), as has been seen in some recent works (*e.g.* Patil et al. 2023). For the MIMIC tasks with larger inputs, ChatGPT (GPT-4) performs best. In fact, GPT-3.5 performs very poorly on ICD-code

**Medication status output**
- aspirin: discontinued
- ibuprofen: neither
- Naprosyn: neither
- Tylenol: active
- Plavix: active

**Evidence highlighting**

Her aspirin (81 mg q.d.) is discontinued, and the patient is advised that she needs to avoid ibuprofen, Naprosyn, alcohol, caffeine, and chocolate. She is advised that Tylenol 325 mg or Tylenol ES (500 mg) is safe to take at 1 or 2 q.4–6h. p.r.n. for pain or fever. Discharge activity is without restriction. DISCHARGE MEDICATIONS: 1. Plavix 75 mg p.o. q.d.

*Figure 2.* Example output and interpretation for medication status. For each element of the output list, our pipeline outputs the text span which contains evidence for that generated output (shown with highlighting).

extraction, perhaps because the task requires not only extracting diagnoses from the input text but also knowing the mapping between diagnoses and ICD codes.

Table 3 contains ablations showing how the different self-verification modules affect the results. The *Omission* step finds missing elements, which increases recall but at the cost of decreased precision.[2] In contrast, the *Prune* step (that incorporates the span from the *Evidence* step) removes extraneous elements, thereby increasing precision. Together (*Full SV*), the steps achieve a balance which improves F1. For tasks with longer inputs (*e.g.* MIMIC-IV ICD-10), the *Omission* step seems to provide more of the improvement in F1, likely because it is able to find evidence that was missed by a single extraction step.

### 3.2. Self-verification yields interpretations

Fig. 2 shows an example output from the self-verification pipeline for medication status (the underlying model is ChatGPT (GPT-4)). In the example, the pipeline correctly identifies each medication and its corresponding status. In addition, the pipeline supplies the span of text which serves as evidence for each returned medication (shown with highlighting). This highlighting enables efficient auditing by a human for each element. In a human-in-the-loop setting, a human could also see results/highlights for elements which were pruned, to quickly check for any mistakes.

Table 4 evaluates the evidence spans provided by SV against

---

[2]It is possible that some false positives identified by the *Omission* step are in fact correct, as the the ICD codes in the MIMIC dataset generally reflect top codes that were billed for rather than all that were present in a note.

*Table 1.* Tasks and associated datasets studied here.

| Task | Data | Example output |
|---|---|---|
| ICD code extraction (ICD-9 and ICD-10) | 250 MIMIC III reports (Johnson et al., 2016), 250 MIMIC IV discharge summaries (Johnson et al., 2021) | [205.0, 724.1, 96.04] |
| Clinical trial arm extraction | 100 annotations to EBM-NLP abstracts (Nye et al., 2018) | [propofol, droperidol, placebo] |
| Medication status extraction | 105 Annotations (Agrawal et al., 2022) to snippets from CASI (Moon et al., 2012) | {aspirin: discontinued, plavix: active} |

*Table 2.* F1 scores for extraction with and without self-verification (SV). Across different models and tasks, SV consistently provides a performance improvement, although it is sometimes small. Bolding shows SV compared to original, underline shows best model for each task. Averaged over 3 random seeds; error bars show the standard error of the mean.

| | ChatGPT (GPT-3.5) | ChatGPT (GPT-4) | GPT-3.5 |
|---|---|---|---|
| Clinical trial arm (Original / SV) | $0.342 \pm 0.010$ / **$0.456 \pm 0.007$** | $0.419 \pm 0.008$ / **$0.530 \pm 0.010$** | $0.512 \pm 0.009$ / **$0.575 \pm 0.003$** |
| Medication name (Original / SV) | $0.892 \pm 0.004$ / **$0.898 \pm 0.002$** | $0.884 \pm 0.003$ / **$0.910 \pm 0.001$** | $0.929 \pm 0.002$ / **$0.935 \pm 0.001$** |
| MIMIC-III ICD-9 (Original / SV) | $0.593 \pm 0.003$ / **$0.619 \pm 0.005$** | $0.652 \pm 0.02$ / **$0.678 \pm 0.007$** | $0.431 \pm 0.03$ / **$0.435 \pm 0.01$** |
| MIMIC-IV ICD-9 (Original / SV) | $0.693 \pm 0.04$ / **$0.713 \pm 0.005$** | $0.718 \pm 0.03$ / **$0.755 \pm 0.004$** | $0.691 \pm 0.02$ / **$0.702 \pm 0.02$** |
| MIMIC-IV ICD-10 (Original / SV) | $0.448 \pm 0.04$ / **$0.464 \pm 0.003$** | $0.487 \pm 0.02$ / **$0.533 \pm 0.002$** | $0.434 \pm 0.03$ / **$0.442 \pm 0.01$** |

*Table 3.* Ablation results when using different combinations of self-verification steps for two tasks. Omission increases Recall and Prune increases Precision. Together they increase both, improving F1. Evidence improves F1 for Medication Status. Underlying model is the best model for each task (GPT-3.5 for *Medication name* and Chat-GPT (GPT-4) for *MIMIC-IV ICD-10*). Averaged over 3 random seeds; error bars are standard error of the mean.

| Medication name | | | |
|---|---|---|---|
| | F1 | Precision | Recall |
| Original | $0.929 \pm 0.002$ | $0.929 \pm 0.003$ | $0.928 \pm 0.003$ |
| + Omission | $0.913 \pm 0.001$ | $0.881 \pm 0.003$ | **$0.946 \pm 0.001$** |
| + Prune | $0.932 \pm 0.002$ | **$0.949 \pm 0.002$** | $0.916 \pm 0.003$ |
| + Full SV | **$0.935 \pm 0.001$** | $0.942 \pm 0.002$ | $0.928 \pm 0.001$ |

| MIMIC-IV ICD-10 | | | |
|---|---|---|---|
| | F1 | Precicion | Recall |
| Original | $0.487 \pm 0.002$ | $0.544 \pm 0.003$ | $0.448 \pm 0.002$ |
| + Omission | $0.517 \pm 0.003$ | $0.553 \pm 0.003$ | **$0.501 \pm 0.004$** |
| + Prune | $0.504 \pm 0.004$ | **$0.557 \pm 0.005$** | $0.451 \pm 0.003$ |
| + Full SV | **$0.533 \pm 0.002$** | **$0.558 \pm 0.002$** | $0.498 \pm 0.002$ |

*Table 4.* Evaluating evidence spans provided by the self-verification pipeline with human-annotated spans. Averaged over 3 random seeds; error bars show standard error of the mean.

| | Span overlap accuracy | Span length |
|---|---|---|
| ChatGPT (GPT-4) | $0.93 \pm 0.02$ | $8.20 \pm 0.48$ |
| GPT-3.5 | $0.84 \pm 0.03$ | $7.33 \pm 0.47$ |

## 4. Discussion

Self-verification constitutes an important step towards unlocking the potential of LLMs in healthcare settings. As LLMs continue to generally improve in performance, clinical information extraction with LLMs + SV seems likely to improve as well.

One limitation of SV is that it incurs a high computational cost as multiple LLM calls are chained together; however, these costs may continue to decrease as models become more efficient (Dao et al., 2022). Another limitation is that LLMs and SV continue to be sensitive to prompts, increasing the need for methods to make LLMs more amenable to prompting (Ouyang et al., 2022; Scheurer et al., 2023) and to make finding strong prompts easier (Shin et al., 2020; Xu et al., 2023; Singh et al., 2022b).

Finally, SV can be harnessed in a variety of ways to improve clinical NLP beyond what is studied here, *e.g.* for studying clinical decision rules (Kornblith et al., 2022), clinical decision support systems (Liu et al., 2023), or improving model distillation (Wu et al., 2023; Toma et al., 2023).

human judgements collected in prior work (Nye et al., 2018). Human reviewers annotated spans in the original text which correspond to interventions, which include clinical trial arms as a subset. Table 4 gives the fraction of generated evidence spans that overlap with a span provided by the human annotators. The fraction is quite large, *e.g.* 93% for ChatGPT (GPT-4). At baseline, human annotators identify less than 3.7% of tokens as interventions, so these span overlap accuracies are much higher than expected by random chance.

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

# A. Appendix

## A.1. Dataset details

**MIMIC**   To preprocess MIMIC data, we follow the steps used by  (Edin et al., 2023). For MIMIC IV, we use the available discharge summaries for each patient while we retrieve more relevant sections from other types of clinical notes for MIMIC III. See the code on Github for complete details.

During LLM extraction, we find that directly extracting ICD codes with an LLM is difficult. Instead, we use the LLM to extract diagnoses, and then postprocess them at the end by asking the LLM to convert each diagnosis to its corresponding ICD code.

**Clinical trial arm dataset**   We manually annotate the clinical trial arms from the first 100 abstract in EBM-NLP (Nye et al., 2018) without the use of any LLMs. All annotations are made available on Github. The mean number of extracted clinical trial arms is 2.14, the maximum is 5 and the minimum is 1.

## A.2. Extended extraction results

*Table A5.* F1 scores for two tasks extracted using a single prompt which concatenates all steps in the SV pipeline. Results are slightly worse than the original extraction presented in Table 2. The prompt contains a paragraph similar to the following: *Before you provide your final response:\n(1) Find any medications in the patient note that were missed.\n(2) Find evidence for each medication as a text span in the input.\nn(3) Verify whether each extracted medication is actually a medication and that its status is correct.* Averaged over 3 random seeds; error bars are standard error of the mean.

|  | ChatGPT (GPT-3.5) | ChatGPT (GPT-4) | GPT-3.5 |
|---|---|---|---|
| Clinical trial arm, original | 0.316 $\pm$0.006 | 0.420 $\pm$0.009 | 0.436 $\pm$0.008 |
| Medication name, original | 0.758 $\pm$0.003 | 0.850 $\pm$0.016 | 0.913 $\pm$0.002 |

