# OpenReview forum: "Self-verification improves few-shot clinical information extraction"
_ICML.cc/2023/Workshop/IMLH — IMLH 2023 PosterShortPaper_

### Official Review · Reviewer_S87c · 2023-06-05
**Self-Verification Improves Few-Shot Clinical Information Extraction**

**Rating:** 8
**Confidence:** 5

**Review:**

The authors propose a self-verification framework to improve the performance of the large language models (LLMs) on few-shot clinical information extraction. Experimental results show that the proposed method can improve the accuracy of the LLMs across multiple clinical information extraction tasks. Additionally, self-verification also yields interpretations for each output.
1.	The proposed self-verification methods consist of four steps: original extraction, omission, evidence and prune steps. The design of prompts is very important for these steps. Although the authors provide the exact prompts in the Github repo, some prompt examples should be added in the main text.
2.	The names of the LLMs are inaccuracy. “text-davinci-003” should be GPT-3, “gpt-3.5-turbo” should be ChatGPT(GPT-3.5), and “gpt4-0314” should be ChatGPT(GPT-4).

---

### Official Review · Reviewer_wGBS · 2023-06-17
**Introduces self-verification for medical information extraction. Proposes important pipeline. Relies on prompt engineering. Only marginal improvements. Short Paper**

**Rating:** 5
**Confidence:** 4

**Review:**

This paper proposes a pipeline for information extraction from medical records by repeatedly making calls to the same LLM. I like the four steps proposed: 1) raw information extraction, 2) omission step to find what values are missing (this is done many times), 3) actually grounding the results by finding evidence in the text, and 4) removing inaccurate results.
However, to achieve these, they rely on creating the right prompts, which is not clear how to do, and there are not enough details in the paper regarding the same. The paper mentions prompts are in the GitHub repo, and it took some time to find the prompts as there was no readme. More importantly, the improvement in results is marginal, which is surprising given how many calls we are making to the LLM. I agree that through SV, we could have improvements in interpretability as measured through the span length. In the current version, SV is compared to human annotators. However, it is very important to compare it with just the original extraction step alone, i.e., what is the span length obtained by LLM without SV?. I really encourage the authors to provide more hypotheses/reasons why the improvements are nominal and consider more automatic prompt learning strategies, such as the one mentioned in https://arxiv.org/abs/2203.05557, so that these methods are extendable to non-English language domains as well.
Overall, this paper presents a principled approach for information extraction from medical records. However, in order to enhance its contribution, it is crucial to address the aforementioned concerns and provide additional justifications for the observed limited improvements.

---

### Meta-Review · Area_Chair_YHEB · 2023-06-20

**Recommendation:** Accept (Poster)
**Confidence:** 5

**Metareview:**

The paper presents a novel pipeline for information extraction from medical records, utilizing a large language model (LLM) integrated with a self-verification framework. Reviewers have acknowledged the proposed four-step approach and have emphasized the significance of prompt design. However, they suggest including additional details and examples of prompts in the main text, as well as comparing the performance of the original extraction step alone. Additionally, they recommend exploring automatic prompt learning strategies to further enhance the system. The limited observed improvements should also be adequately justified in the final version. On the whole, the paper is deemed interesting and offers a comprehensive and principled approach to address this crucial clinical task. The reviewing committee recommends accepting the paper for publication.

---

### Decision · Program_Chairs · 2023-06-20

Accept (Poster Short Paper)